# Emotional Intelligence in Hemodialysis Patients: The Impact of an Intradialytic Exercise Training Program

**DOI:** 10.3390/healthcare12090872

**Published:** 2024-04-23

**Authors:** Stefania S. Grigoriou, Christina Karatzaferi, Christoforos D. Giannaki, Giorgos K. Sakkas

**Affiliations:** 1School of Physical Education, Sport Science and Dietetics, University of Thessaly, 38221 Trikala, Greece; stefania.grigoriou@lvr.de (S.S.G.); ck@uth.gr (C.K.); 2Department of Life Sciences, University of Nicosia, Nicosia 2417, Cyprus; giannaki.c@unic.ac.cy; 3Research Centre for Exercise and Nutrition (RECEN), University of Nicosia, Nicosia 2417, Cyprus; 4School of Sports and Health Sciences, Cardiff Metropolitan University, Cardiff CF5 2YB, UK

**Keywords:** emotional intelligence, fatigue, intradialytic exercise, quality of life

## Abstract

The current study aimed to investigate whether there is a relationship between emotional intelligence (EI), functional capacity, fatigue, cognitive function, and quality of life (QoL) in HD patients and to assess the effect of a 9-month intradialytic exercise training program on EI levels. Seventy-eight dialysis patients (50 M/28 F, 60.6 ± 17.2 years) participated in the cross-sectional study. Afterward, a subgroup of 18 patients (15 M/3 F, 56.7 ± 12.3 years) completed a 9-month supervised intradialytic exercise training program (three times weekly). EI was assessed by the Schutte Self Report Emotional Intelligence Test (SSEIT) and the Wong and Law Emotional Intelligence Scale (WLEIS). Functional capacity was assessed by a battery of tests. Sleep quality, depression levels, and daily sleepiness were assessed via validated questionnaires. All assessments were carried out before and after the intervention. A significant positive correlation was found between the WLEIS scores and the physical component summary of the QoL questionnaire. In contrast, the WLEIS scores were negatively associated with general and physical fatigue. The SSEIT scores were positively associated with cognitive function. After nine months of exercise training, only the group with low WLEIS scores improved their EI score significantly compared to the baseline values (98.7 ± 7.0 vs. 73.0 ± 4.0, *p* = 0.020), while no changes were observed in the medium or high EI groups. In conclusion, patients with higher levels of EI showed increased quality of life and lower levels of fatigue. Patients with low levels of EI are more likely to benefit from an exercise training program compared to their medium- and high-level counterparts.

## 1. Introduction

Patients with end-stage renal disease (ESRD) may receive hemodialysis (HD) therapy as a maintenance treatment, which is considered both a life-saving and life-altering process. The disease per se, along with HD therapy, induces serious alterations in various physical, social, and mental-related parameters, which can negatively impact the patient’s quality of life (QoL). More specifically, it has been shown that patients receiving HD therapy experience very low levels of QoL, which are usually accompanied by significant emotional distress symptoms such as depression and anxiety [1]. Additionally, low QoL scores are strongly correlated with higher levels of hospitalization and mortality and accordingly are considered an established predictor of these two factors [2,3,4]. Indeed, low levels of QoL are associated with the patient’s mental, social, and physical problems [1,5,6], including depression [7,8]. 

Emotional intelligence (EI) refers to a set of social cognitive abilities specifically related to emotions. EI has been conceptualized as “the ability to engage in sophisticated information processing about one’s own and others’ emotions and the ability to use this information as a guide to thinking and behavior” [9]. Research shows that high levels of EI are negatively related to anxiety and depression symptoms [10]. Moreover, fatigue, sleep disturbances, anxiety, and depression, which are common problems in patients on HD [11], are associated with EI in healthy participants [12]. Recent research suggests an important relationship between EI and health-related QoL in patients with chronic diseases [13]. A study by Rey et al. [13] examined EI concerning personality and health-related QoL in cancer patients and found that EI was positively associated with different dimensions of health-related QoL. The authors proposed that EI could be a useful variable to assess and identify patients who may be at risk for experiencing low health-related QoL [13].

So far, there is limited information relating to EI levels in HD patients. The study by Khan et al. [14] published in 1971 examined the emotional status, level of intelligence, and self-concept in children with CKD and found that the highest percentage of the children suffered from serious social and emotional difficulties, such as feelings of social isolation, excessive dependency upon their parents, and depression [14]. To our knowledge, only a few studies have examined the relationships between EI, quality of life, and other psychological variables in HD patients. In a recent study in HD patients, EI was found to be significantly related to post-traumatic growth and, thus, mediates the relationship between uncertainty, anxiety, and depression [15]. According to the current literature, high physical activity levels are associated with good levels of EI in healthy individuals [16,17], while it is still unknown whether exercise can improve EI in hemodialysis patients.

The primary aim of the cross-sectional part of this study was to investigate whether there are relationships between emotional intelligence and aspects linked to health-related QoL in HD patients. The secondary aim was to investigate whether an exercise intervention training program could alter EI levels.

## 2. Materials and Methods

Seventy-eight dialysis patients (50 M/28 F, 61.2 ± 17.1 years) were recruited from local HD units in order to participate in the cross-sectional study. Patients were divided into groups according to their scores on the EI scales. As two different scales of EI (WLEIS and SSEIT) were used to assess EI levels in the current study, the 78 patients were divided into two groups per scale (low WLEIS and high WLEIS, and low SSEIT and high SSEIT) using the quartiles approach. Therefore, four patient subgroups were used. In addition, the rationale behind the development of 4 patient subgroups based on WLEIS and SSEIT scores is to assess whether the level of EI (low or high score) could affect the effectiveness of an exercise training program. The four patient subgroups were developed using the quartile approach, using the lower quartile for the low group and the upper quartile for the high group, excluding the values for the median quartile. For the WLEIS, the lower quartile of patients scored less than 79, and the upper quartile scored greater than 96. For the SSEIT, the lower quartile of patients scored less than 111, and the upper quartile scored greater than 134.

The 78 patients were also assessed for eligibility in order to participate in a 9-month exercise training program (Figure 1). Eighteen stable HD patients (15 M/3 F, 56.7 ± 12.3) completed the 9-month supervised exercise training program during HD (3 times weekly). The exercise sessions were supervised by 2 specialized exercise scientists. The 18 patients were divided into 3 groups according to the WLEIS score (low, medium, and high) for further analysis.

Inclusion criteria were dialysis for at least six months with adequate dialysis delivery (Kt/V > 1.1) and stable clinical condition. Exclusion criteria were as follows: being unable to give informed consent; opportunistic infection within 3 months prior to enrollment; malignancy or infection requiring intravenous antibiotics within 2 months prior to enrollment; HIV, musculoskeletal contraindication to exercise, or requirement for systemic anticoagulation; participating or having participated in an investigational drug or medical device study within 30 days or five half-lives; females who were pregnant, breast feeding or of childbearing potential who did not agree to remain abstinent or to use an acceptable contraceptive regimen; patients who were judged to have clinically-significant abnormalities upon clinical examination or laboratory testing; patients who were unable to adequately cooperate because of personal or family conditions; or patients who suffered from a mental disorder that interferes with the diagnosis and/or with the conduct of the study (e.g., patients were excluded if they had been diagnosed with schizophrenia, major depression, or dementia).

The study was approved by the Human Research and Ethics Committee of the University of Thessaly, the Bioethics Committee of the University General Hospital of Larissa, and the General Hospital of Trikala, Greece. All patients gave their written informed consent prior to study participation. The whole study was registered at ClinicalTrials.gov (NCT01721551) as a clinical trial, while this current study presents a subset of data acquired under the registered RCT study.

### 2.1. HD Procedure

The patients underwent HD therapy (Fresenius Medical Care, Fresenius 4008B, Oberursel, Germany) 3 times per week with low flux, hollow-fiber dialyzers and a bicarbonate buffer. The HD session lasted 4 h. An enoxaparin dose of 40–60 mg was administered intravenously before the beginning of each HD session. EPO therapy was given after the completion of the HD session in order to normalize hemoglobin levels within 11–12 g/dL.

### 2.2. General Study Design

Patients were assessed before (PRE) and after (POST) the 9-month aerobic exercise training program. Exercise training was implemented during HD while the exercise program was supervised by 2 specialized exercise trainers. A cycle exercise was performed 3 times weekly for 60 min each time, starting between the first 2 h of HD using an adapted cycle ergometer (Model 881 Monark Rehab Trainer, Monark, Varberg, Sweden) at an intensity of 50–60% of the patient’s maximal exercise capacity, which was estimated during a previous HD session [18,19]. During and before release from the HD unit, body mass, systolic and diastolic blood pressures (SBP, DBP), and heart rate (using the RS800CX, Polar Electro Oy, Kempele, Finland) were monitored and recorded. Participants’ blood chemistry records were recorded before and at the end of the 9-month study. Participants were assessed in aspects related to mental and physical health as well as exercise and functional capacity.

### 2.3. Emotional Intelligence

Emotional intelligence was assessed by two tests. Firstly, the Schutte Self Report Emotional Intelligence Test (SSEIT) [20] was used. This is a 33-item self-report instrument where patients are asked to indicate their responses to items reflecting adaptive tendencies towards emotional intelligence according to a 5-point scale, with “1” representing strong agreement and “5” representing strong disagreement.

Secondly, the Wong and Law Emotional Intelligence Scale (WLEIS) [21], which is a shorter instrument and is administered by an external examiner, was also completed by the participants. This test contains 16 items grouped into four subscales as follows: (a) self-emotion appraisal (SEA), (b) emotion appraisal of others (OEA), (c) use of emotion (UOE), and (d) regulation of emotion (ROE).

### 2.4. Exercise Capacity

Using an incremental cycle ergometer test [19], exercise capacity was assessed before, at 3 months, at 6 months, and at the end of the 9-month exercise intervention. Values recorded were used to re-adjust the submaximal training intensity of the intradialytic exercise sessions of this program.

### 2.5. Functional Capacity and Strength

The patient’s functional capacity levels were evaluated using two sit-to-stand functional tests from which three scores were recorded (time taken to complete 5 sit-to-stands [STS-5], number of repetitions in 30 s [STS-30], and number of repetitions in a whole minute [STS-60]). Maximum isometric handgrip strength (HGS) was measured on the non-fistula (dominant) side [22] using a handgrip dynamometer (Charder MG4800 Medical Handgrip Dynamometer, Charder Electronic Taiwan, Guozhong Rd., Dali Dist., Taichung City, 41262 Taiwan (R.O.C.)). 

### 2.6. Body Composition

Each patient’s dry weight (ideal weight after removal of excess fluids) was recorded. Dry weight and height were used to calculate body mass index (BMI). Waist and hip peripheries were measured and the waist-to-hip ratio (WHR) was calculated. Body composition was assessed using a whole-body, multi-frequency, bio-impedance spectroscopy system (BCM^®^, Fresenius Medical Care, Bad Homburg, Germany) to estimate fat mass (FM), lean tissue mass (LTM), total body water (TBW), and body cell mass (BCM) [23]. The body composition measurements were taken immediately before the initiation and after the completion of the HD session, with the participants resting in the supine position. Electrodes were placed on the wrist of the arm without the arterio-venous fistula as well as on the ipsilateral ankle and connected to the BCM device [24]. 

## 3. Questionnaires

### 3.1. Fatigue

Fatigue was estimated using various questionnaires evaluating chronic and acute aspects of fatigue. The general severity of fatigue was assessed using the Fatigue Severity Scale (FSS) [25]. This questionnaire contains nine statements concerning respondents’ fatigue. Subacute fatigue was assessed using the Brief Fatigue Inventory (BFI) [26], which is an instrument that can be administered in a clinical setting to assess the severity of fatigue experienced by patients, as well as its impact on their ability to function over the previous 24 h. Finally, the Multidimensional Fatigue Inventory (MFI) [27], which is a 20-item scale designed to evaluate the dimensions of general fatigue, physical fatigue, reduced motivation, reduced activity, and mental fatigue, was completed by participants.

### 3.2. Cognitive Function

Cognitive function was assessed by the Mini-Mental State Examination questionnaire (MMSE) [28], which is a brief 30-point questionnaire test that evaluates cognitive impairment. This questionnaire consists of simple questions and problems in a number of areas: the time and place of the test, repeating lists of words, arithmetic such as the serial sevens, language use and comprehension, and basic motor skills.

## 4. Symptoms of Depression

Depressive symptoms were evaluated using the Zung Self-Rating Depression Scale (with a score > 44 being considered the cut-off for diagnosis of depression). Moreover, the Beck Depression Inventory II (Beck Depression Test, BDT) [29] was used to assess the intensity of depressive symptoms. 

### 4.1. Pain Perception

Each participant also completed the Fibromyalgia Impact Questionnaire (FIQ) [30]. This self-administered questionnaire was developed to measure fibromyalgia (FM) patient status, progress, and outcomes. The instrument contains 11 questions measuring physical functioning, work status (missed days of work and job difficulty), depression, anxiety, morning tiredness, pain, stiffness, fatigue, and well-being over the past week. 

### 4.2. Perceived Quality of Life

Quality of life was assessed by the Generic Medical Outcomes 36-Item Short Form Survey (SF-36) [31], which contains eight dimensions, generating a profile of health-related quality of life. These dimensions are (1) physical functioning; (2) role limitations due to physical functioning; (3) bodily pain; (4) general health perceptions; (5) vitality; (6) social functioning; (7) role limitations due to emotional functioning; and (8) mental health. Total SF-36 QoL scores range from 0 (extremely poor) to 100 (very good). Moreover, the quality of life in patients was evaluated by the Missoula–VITAS Quality of Life Index Version-15R. The MVQOLI is an assessment instrument that gathers patient-reported information about QoL during advanced illness. We used the short version with 15 items, which is for HD patients [32]. 

### 4.3. Sleep Quality and Daily Sleepiness

Sleep disturbances and usual sleep habits during the preceding month were evaluated by the Pittsburgh Sleep Quality Index (PSQI), which contains 19 questions [33]. 

Furthermore, the HD patients’ daily sleepiness status was assessed by the Epworth Sleepiness Scale (ESS) [34]. This scale differentiates between typical sleepiness and excessive daytime sleepiness that requires intervention. 

### 4.4. Statistical Analysis

Continuous variables were analyzed using independent sample *t*-tests. In the case of outcome variables that changed in the same direction in both the progressive exercise and control groups, between-group comparisons were also made (comparing Δ-change values) to evaluate if the change in one group was significantly greater than that of the other group. Kendall’s correlation test was used to assess the relationships between the examined variables. For comparing initial and final values (pre- and post-exercise training), two-way, repeated measures analysis of variance tests (ANOVAs) were performed. All statistical analyses were performed using SPSS version 21.0 (SPSS Inc. Chicago, IL, USA). Data are presented as mean ± SD, and the level for statistical significance was set at *p* ≤ 0.05. 

## 5. Results

Table 1 shows the baseline characteristics of the participants. All the patients successfully completed the questionnaires, and no adverse effects were reported.

The results of the correlations between patients’ characteristics and functional capacity are presented in Table 2. 

A *t*-test was conducted in order to determine if significant differences in EI scores between divided groups existed: low WLEIS, low SSEIT, high WLEIS, and high SSEIT. The EI data are presented in Table 3. There were no significant differences between the groups in the functional capacity tests performed. There were significant differences in the scores of the Mini-Mental State Exam (MMSE) and quality of life (MVQOLI) between the WLEIS group patients. 

There was a statistically significant difference between the general perceptions of fatigue scores (FSS, BFI) and divided groups of EI regarding the WLEIS scores (Table 3). Furthermore, it was observed that patients with lower EI scores showed higher scores in general fatigue.

Taking into account the differences in the questionnaire scores regarding the EI status in the HD patients’ groups, MMSE scores increased in the group with high scores in the SSEIT [M (high EI SSEIT) = 26.8, SD = 2.5, *p* = 0.046], while the group with low SSEIT scores had low scores in the MMSE [M (low EI SSEIT) = 25.0, SD = 2.9, *p* = 0.046]. In addition, there was a statistically significant difference between the general perceptions of fatigue scores (FSS, MFI) and divided groups of EI. Patients with low EI status showed higher scores in general fatigue. Specifically, the FSS score was higher in the group with a low SSEIT score [M (low EI SSEIT) = 5.1, SD = 2.6, *p* = 0.015] than in the group with a high SSEIT score [M (high EI SSEIT) = 3.9, SD = 1.8, *p* = 0.015]. Furthermore, the MFI score still was correspondingly high in the low SSEIT group [M (low EI SSEIT) = 61.9, SD = 19.1, *p* = 0.008] compared to the high SSEIT group [M (high EI SSEIT) = 47.9, SD = 16.6, *p* = 0.008]. 

The subset of 18 patients completed the exercise intervention. Their mean age was 56.7 ± 12.3 years (15 males), and their mean duration in HD treatment was 61.7 ± 47.5 months. The effects of the 9-month exercise training on emotional intelligence are presented in Table 4. 

Two-way repeated measures analyses of variance (ANOVAs) were conducted to determine if there were significant differences in the WLEIS EI test between the divided groups: low WLEIS, medium WLEIS, and high WLEIS. The EI data are presented in Table 5. 

## 6. Discussion

This study initially investigated whether there is a relationship between EI, functional capacity, cognitive function, fatigue, and quality of life in HD patients and whether an exercise training program could change the level of EI in patients receiving HD therapy. The results of the cross-sectional part of this study revealed significant correlations between EI and quality of life, fatigue, and cognitive function in HD patients. Secondly, a nine-month intradialytic exercise training program was applied; however, it was effective only in patients with low EI, while it was ineffective in terms of inducing changes in EI in patients with moderate to high EI scores.

The relationship between EI status and quality of life has been investigated in other patients with chronic diseases reporting similar results including cancer [13,35] and chronic obstructive pulmonary disease (COPD) patients [36]. According to the above authors, EI is an ability that can be learned and could be a complementary, non-expensive tool to the rehabilitation programs aiming to improve the low levels of QoL that patients with chronic diseases experience.

Furthermore, an association between EI and fatigue in HD patients was found in this study. Studies have shown that a higher EI status is associated with less fatigue in healthy participants [12]. Specifically, Brown and Schutte who assessed EI status in university students in combination with the psychosocial variables of depression, anxiety, optimism, internal health locus of control, amount of social support, and satisfaction with social support, indicated that EI and fatigue had an impact on each variable [12]. Indeed, previous studies that examined the relationship between EI and depression revealed a significant negative relationship between EI and depression [37]. Specifically, Downey et al. (2008) investigated whether an association exists between EI and clinical diagnosis of depression and demonstrated that measurement of EI has a predictive value for assessing patients with a high risk of developing depression [37]. Furthermore, the negative relationship between EI and depression has been observed among adolescents [38]. Specifically, a recent study by Balluerka et al. demonstrated that high levels of emotional clarity and repair were related to lower levels of depressed mood in adolescents [39]. In addition, another study by Vlachaki et al. (2013), who examined the relationship between different dimensions of EI and coronary heart disease, found that facets of trait EI were associated with a high incidence of coronary heart disease [40].

A correlation between EI and cognitive function was found in this study, indicating that patients with higher EI scores had higher scores on the MMSE. Previous findings from other studies have supported that people with higher EI are likely to achieve higher scores on a cognitive task [41] Moreover, high EI individuals confronted with difficulties in a cognitive task were able to ward off the detrimental emotional effects and persist on the task [41].

Nine months of exercise training improved the EI status of the HD patients with low WLEIS but not in the whole sample. The failure of the examined exercise intervention to improve the EI status of the whole sample could be attributed to the fact that the exercise intensity was moderate, between 50 and 60% of the patient’s maximal exercise capacity, while a higher level of intensity could have been more effective. According to published guidelines on exercise in hemodialysis patients, aerobic exercise at an intensity above 60% of the patients’ maximum capacity is recommended to improve various physiological and health-related parameters [42], and this is the reason why we chose the intensity for the current study. However, we are aware that this level of intensity is higher [43] or lower [44] compared to the levels utilized in similar studies within the same population. 

Regarding the association between EI and physical activity, a contemporary review by Grigoriou et al. (2012) demonstrated that EI is positively related to good health and exercise habits. Specifically, participation in vigorous and moderate physical activities seems to positively affect EI [45]. Solansky and Lane (2010) have supported that exercise beneficially regulates mood, and this mood improvement had a positive impact on EI status. These authors suggested that exercise training could improve mood and accordingly increase EI scores [46]. EI could be developed and learned at any time or age and, in combination with exercise training programs, could increase well-being and better emotional regulation in patients with chronic diseases [36]. 

Cognitive impairment is common in chronic kidney disease patients, particularly among those receiving hemodialysis therapy [47,48]. Although exercise intervention was found to be beneficial specifically for patients with low EI levels in the present study, prior research has demonstrated the effectiveness of intradialytic exercise in enhancing cognitive function [49]. The increased cerebral blood flow resulting from exercise has been identified as a potential mechanism through which intradialytic aerobic exercise may help reduce cognitive decline and preserve brain function [50]. Other potential mechanisms include improved sleep, reduced depression, and improved cardiovascular-related parameters, all reported to be improved after intradialytic aerobic exercise training in HD populations [51,52]. Unfortunately, the investigation of the physiological mechanisms that could explain the beneficial effects of exercise on EI and cognition was out of the scope of this trial. Emotional intelligence is a promising parameter for influencing physical and psychological characteristics in populations suffering from chronic conditions or diseases. It seems that exercise, especially chronic exercise training, can alter the state of EI, improving the mental state and facilitating changes in the physical component. However, more research is necessary to assess the precise exercise prescription needed to maximize the positive effect on human health. 

There are some limitations in the current study that need to be acknowledged. The small pool of subjects that completed the exercise intervention (N = 3 in one of the subgroups (Low EI)) and the lack of a control group could have jeopardized the generalization of the findings of the current study. In addition, the cohort of patients was not gender balanced as can be seen in both the cross-sectional and intervention parts of the study. Finally, as the intervention part of the study used a small number of patients, the results should be treated with caution. 

## 7. Conclusions

In conclusion, this is the first study to investigate EI levels in HD patients using two different EI questionnaires and assessing the impact of an intradialytic exercise training program. There was a significant positive correlation between EI scores and physical fitness and cognitive function, while it was negatively associated with fatigue. An intradialytic exercise training program lasting for 9 months improved EI scores, especially in the group with low EI scores. A better understanding of the interactions between factors that influence EI and functional capacity may help researchers develop interventions for quality-of-life improvement among dialysis patients. Even though the EI is a construct that has not been recognized yet as a useful tool in healthcare, future research is needed to focus on healthcare aspects. 

## Figures and Tables

**Figure 1 healthcare-12-00872-f001:**
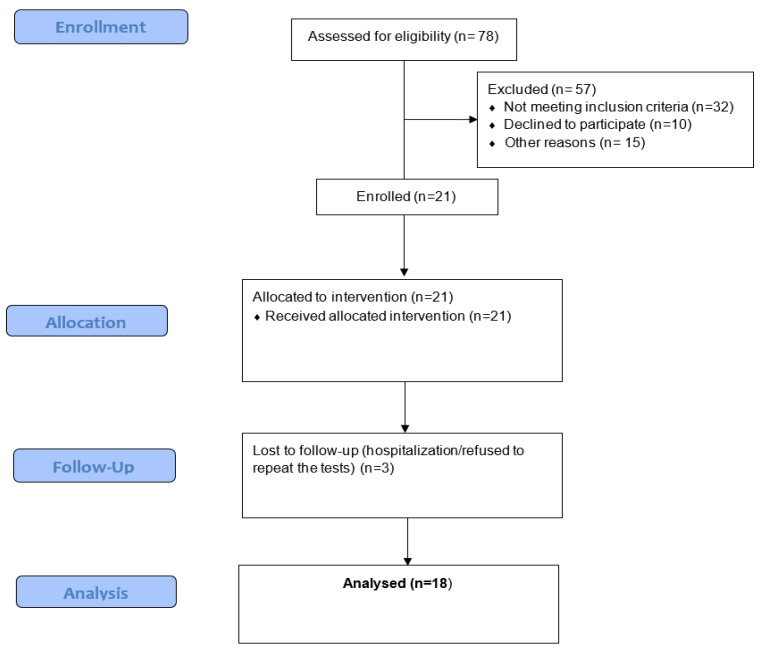
Flow of participants through the exercise intervention study.

**Table 1 healthcare-12-00872-t001:** Patients’ basic characteristics, functional capacity, and questionnaire assessment.

Variables	Pool Data	Pre-Training Group	Post-Training Group	*p* Value *
Μ/F	50/28	15/3	15/3	
Age	61.2 ± 17.1	56.7 ± 12.3	56.7 ± 12.3	
BMI	25.8 ± 4.9	25.4 ± 4.4	25.9 ± 4.7	0.080
WHR	1.0 ± 0.1	1.0 ± 0.1	1.0 ± 0.1	0.914
Pills/day	7.7 ± 4.7	8.0 ± 5.0	7.6 ± 4.0	0.270
Months in dialysis	63.4 ± 67.0	52.7 ± 47.5	61.7 ± 47.5	
MMSE	25.3 ± 2.9	26.3 ± 1.9	27.0 ± 2.0	0.010
FSS	4.9 ± 2.4	3.6 ± 1.3	3.6 ± 1.6	0.989
FIQ	22.5 ± 16.1	14.9 ± 4.6	7.4 ± 5.7	0.000
WLEIS	81.9 ± 22.7	88.4 ± 9.3	89.4 ± 12.6	0.802
SSEIT	121.5 ± 19.9	128.8 ± 12.7	133.2 ± 16.3	0.438
SF-36—Physical Health	75.9 ± 89.6	65.9 ± 17.8	66.9 ± 18.1	0.760
SF-36—Mental Health	66.4 ± 66.6	65.0 ± 11.1	65.0 ± 12.3	0.955
SF-36—Total	48.7 ± 21.5	70.0 ± 13.1	67.0 ± 15.8	0.418
BDI	9.5 ± 7.4	5.3 ± 4.9	6.9 ± 6.4	0.195
MVQOL	16.1 ± 4.3	18.3 ± 3.6	18.7 ± 3.2	0.664
ESS	4.5 ± 2.6	4.3 ± 2.6	5.4 ± 2.8	0.328
PSQI	9.3 ± 4.8	5.6 ± 3.0	5.9 ± 4.9	0.771
Pain	1.7 ± 3.6	0.1 ± 0.6	0.1 ± 0.5	0.331
BFI	2.7 ± 2.6	1.5 ± 1.1	1.9 ± 1.8	0.332
Handgrip (Nw)	23.5 ± 10.8 (N = 38)	26.5 ± 8.2	29.4 ± 8.2	0.031
STS 5 (sec)	10.9 ± 4.4 (N = 24)	12.7 ± 5.2	9.6 ± 3.2	0.042
STS30 (rep)	12.1 ± 3.5 (N = 24)	11.5 ± 2.4	16.1 ± 3.7	0.001
STS60 (rep)	22.4 ± 7.1 (N = 24)	20.0 ± 5.2	28.7 ± 7.6	0.001

Abbreviations: BMI, Body Mass Index; WHR, waist-to-hip ratio; MMSE, Mini-Mental State Exam; FSS, Fatigue Severity Scale; FIQ, Fibromyalgia Impact Questionnaire; WLEIS, Wong and Law Emotional Intelligence Scale; SSEIT, Schutte Self-Report Emotional Intelligence Test; BDI, Beck Depression Inventory; MVQOLI, Missoula–VITAS Quality of Life Index; ESS, Epworth Sleepiness Scale; PSQI, Pittsburgh Sleep Quality Index; BFI, Brief Fatigue Inventory; STS, sit-to-stand. * Pre/post comparisons.

**Table 2 healthcare-12-00872-t002:** Correlations between the examined parameters.

**Correlations between Patients’ Basic Characteristics, Functional Capacity, and Emotional Intelligence**							
	**Gender**	**Age**	**BMI**	**WHR**	**Pills/Day**	**Months on HD**	**Handgrip**
WLEIS	r = 0.037	r = −0.710	r = −0.094	r = 0.070	r = −0.190	r = −0.150	r = 0.091
	*p* = 0.740	*p* = 0.450	*p* = 0.350	*p* = 0.642	*p* = 0.098	*p* = 0.195	*p* = 0.443
SSEIT	r = −0.062	r = −0.117	r = −0.007	r = 0.006	r = 0.023	r = 0.037	r = 0.070
	*p* = 0.569	*p* = 0.209	*p* = 0.942	*p* = 0.966	*p* = 0.839	*p* = 0.745	*p* = 0.551
**Correlations between Patients’ Emotional Intelligence and Questionnaires**							
	**MMSE**	**FSS**	**FIQ**	**SF-36 PCS**	**SF-36 MCS**	**WLEIS**	**SSEIT**
WLEIS	r = 0.216	r = −0.215 *	r = −0.023	r = 0.191 *	r = 0.064	-	r = 0.407 **
	*p* = 0.081	*p* = 0.024	*p* = 0.848	*p* = 0.037	*p* = 0.488	-	*p* = 0.000
SSEIT	r = 0.268 *	r = −0.222 *	r = 0.146	r = 0.058	r = 0.050	r = 0.407 **	-
	*p* = 0.029	*p* = 0.018	*p* = 0.218	*p* = 0.524	*p* = 0.579	*p* = 0.000	-
	**BDI**	**MVQOLI**	**ESS**	**PSQI**	**FIQ**	**MFI**	**BFI**
WLEIS	r = −0.109	r = 0.255 **	r = 0.002	r = −0.077	r = 0.104	r = −0.206 *	r = −0.195
	*p* = 0.254	*p* = 0.007	*p* = 0.985	*p* = 0.440	*p* = 0.421	*p* = 0.031	*p* = 0.102
SSEIT	r = −0.140	r = 0.205 *	r = −0.054	r = −0.001	r = 0.145	r = −0.255 **	r = −0.031
	*p* = 0.139	*p* = 0.027	*p* = 0.576	*p* = 0.995	*p* = 0.256	*p* = 0.006	*p* = 0.796

Abbreviations: HD, hemodialysis; BMI, Body Mass Index; WHR, waist-to-hip ratio; MMSE, Mini-Mental State Exam; FSS, Fatigue Severity Scale; FIQ, Fibromyalgia Impact Questionnaire; MCS, mental component summary; PCS, physical component summary; WLEIS, Wong and Law Emotional Intelligence Scale; SSEIT, Schutte Self-Report Emotional Intelligence Test; BDI, Beck Depression Inventory; MVQOLI, Missoula–VITAS Quality of Life Index; ESS, Epworth Sleepiness Scale; PSQI, Pittsburgh Sleep Quality Index; MFI, Multidimensional Fatigue Inventory; BFI, Brief Fatigue Inventory. * *p* < 0.05, ** *p* < 0.01.

**Table 3 healthcare-12-00872-t003:** Basic characteristics, functional capacity, and questionnaire data divided into two groups according to scores in the two emotional intelligence questionnaires.

Variables	Low EI SSEIT	High EI SSEIT	Low EI WLEIS	High EI WLEIS	*p* Value
N	54	16	58	17	
Gender (M/F)	37/17	10/6	41/17	9/8	0.672 0.213
Age	61.3 ± 17.9	56.9 ± 15.1	62.3 ± 16.8	56.7 ± 17.9	0.337 0.259
BMI	26.8 ± 4.9	24.1 ± 4.7	26.3 ± 5.1	24.4 ± 4.2	0.069 0.125
WHR	1.0 ± 0.1	1.0 ± 0.4	0.9 ± 0.1	1.0 ± 0.3	0.648 0.409
ABI	1.0 ± 0.1	1.0 ± 0.2	1.0 ± 0.1	1.0 ± 0.1	0.765 0.337
Months in Dialysis	69.4 ± 76.4	53.6 ± 37.9	72.1 ± 73.7	45.3 ± 45.9	0.338 0.125
Hand Grip (Newton)	23.7 ± 11.1	26.4 ± 10.4	23.8 ± 11.0	26.1 ± 10.6	0.461 0.493
**Questionnaires**					
WLEIS	84.4 ± 13.2	93.4 ± 8.4	82.3 ± 11.7	99.4 ± 2.8	**0.003 **** **0.000 ****
SSEIT	114.7 ± 17.3	143.9 ± 6.8	117.8 ± 20.5	133.6 ± 10.9	**0.000 **** **0.000 ****
MMSE	25.0 ± 2.9	26.8 ± 2.5	25.0 ± 3.1	26.7 ± 1.5	**0.046 *** **0.011 ***
FSS	5.1 ± 2.6	3.9 ± 1.8	5.2 ± 2.5	3.9 ± 1.9	**0.045 *** **0.044 ***
MFI	61.9 ± 19.1	47.9 ± 16.6	60.8 ± 19.0	51.7 ± 18.6	**0.008 **** 0.095
BFI	2.6 ± 2.6	2.5 ± 2.6	3.0 ± 2.7	1.4 ± 1.3	0.912 **0.011 ***
Physical Health (SF-36)	82.0 ± 93.4	55.9 ± 36.8	66.9 ± 80.8	96.7 ± 85.6	0.100 0.213
Mental Health (SF-36)	69.4 ± 74.3	65.4 ± 49.5	70.7 ± 74.9	55.3 ± 24.7	0.805 0.185
MVQOL	16.0 ± 4.2	17.8 ± 4.3	15.6 ± 4.1	18.6 ± 3.8	0.144 **0.013 ***
BDI	9.9 ± 7.5	7.3 ± 7.1	9.8 ± 7.4	7.4 ± 6.7	0.221 0.226
ESS	4.5 ± 2.5	4.3 ± 3.0	4.2 ± 2.5	4.9 ± 2.9	0.839 0.437
PSQI	8.5 ± 5.3	8.6 ± 4.7	8.5 ± 5.0	8.2 ± 5.5	0.940 0.865
FIQ	22.4 ± 15.7	25.2 ± 18.0	24.1 ± 16.9	19.6 ± 13.3	0.653 0.341
FIQ	1.6 ± 3.5	2.4 ± 3.4	1.6 ± 3.5	2.1 ± 4.0	0.512 0.658

Abbreviations: BMI, Body Mass Index; WHR, waist-to-hip ratio; ABI, Ankle Brachial Index; WLEIS, Wong and Law Emotional Intelligence Scale; SSEIT, Schutte Self-Report Emotional Intelligence Test; MMSE, Mini-Mental State Exam; FSS, Fatigue Severity Scale; MFI, Multidimensional Fatigue Inventory; BFI, Brief Fatigue Inventory; MVQOLI, Missoula–VITAS Quality of Life Index; FIQ, Fibromyalgia Impact Questionnaire; BDI, Beck Depression Inventory; ESS, Epworth Sleepiness Scale; PSQI, Pittsburgh Sleep Quality Index; FIQ, Fibromyalgia Impact Questionnaire. * *p* < 0.05, ** *p* < 0.01.

**Table 4 healthcare-12-00872-t004:** Effect of 9 months of exercise training on the emotional intelligence questionnaires.

	Pre-Training	Post-Training	*p* Values
N	18	18	
WLEIS	88.4 ± 9.3	89.4 ± 3.0	0.802
SSEIT	128.8 ± 133.2	133.2 ± 16.3	0.438

Abbreviations: WLEIS, Wong and Law Emotional Intelligence Scale; SSEIT, Schutte Self-Report Emotional Intelligence Test.

**Table 5 healthcare-12-00872-t005:** Effect of 9 months of exercise training on the emotional intelligence questionnaire WLEIS. Data divided into three groups according to scores in the emotional intelligence test.

	Low	Medium	High	*p* Values
N	3	11	4	
Pre-WLEIS score	73.0 ± 4.0	88.6 ± 4.7	99.2 ± 2.6	0.106
Post-WLEIS score	98.7 ± 7.0 *	85.7 ± 14.0	92.5 ± 7.9	0.541
Delta changes	25.7 ± 6.4	−2.9 ± 15.2	−6.7 ± 5.9	0.048

Abbreviations: WLEIS, Wong and Law Emotional Intelligence Scale. *p* values GLM repeated measures; * Pre/post differences, *p* = 0.02.

## Data Availability

Data are unavailable due to privacy or ethical restrictions. They will be available upon request.

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
