# Peer review of "Emotional Intelligence in Hemodialysis Patients: The Impact of an Intradialytic Exercise Training Program"

_healthcare, 2024, doi:10.3390/healthcare12090872_

Round 1

Reviewer 1 Report

Comments and Suggestions for Authors

The their study, the authors investigated the influence of a nine month bicycle ergometer training during hemodialysis sessions 3 times a week on emotional intelligence and health related quality of life.

Initially 78 patients were screened, 23 completed the subsequent 9-month training. Patients with poor initial results in EI tests improved in terms of quality of life and fatique, patients with better initial results did not improve as a consequence of the intervention.

The question posed in this study is very important in the context of the care of patients on hemodialysis and the current issue of the importance of sport on dialysis and covers a new aspect that has not been investigated before. The type of training and the duration of the study seem reasonable.

However, the article raises some questions: 

Methods/Design:

- What was the rationale for choosing the two EI tests? Why were common tests such as the Mayer-Salovey-Caruso Emotional Intelligence Test (MSCEIT) not used?

- What is the rationale for the division into 4 subgroups depending on the test procedure? This should be explained in more detail. How are the results to be evaluated in comparison to the healthy normal population? The authors should address this in the discussion section.

- Why were the tests in the non-intervention group not also repeated after 9 months? Does repetitive testing play a role with these tests?

Results: 

- Table 3: Number of patients for SSEIT and WLEIS does not add up to 78, please explain. 

- Table 5: Do the p-values refer to the differences between the categories? There is comment regarding the significance of the “delta change”. The "Low" category consists of only 3 patients. How does this influence the results? Please discuss.

Discussion:

- The part of the discussion on the influence of chronic diseases on EI and vice versa takes up a long section and could be shortened. 

- Line 350: The statement is contrary to the previous statement that training at low initial values has an influence. 

- Training at 50% of maximal exercise capacity for 1 h is above the usual training levels in other studies. This should be compared, e.g. reference to the recently published DiATT study (NEJM Evid 2023;2(9)). 

- The authors should make a statement to the usually worsening cognitive function in chronic hemodialysis patients (for example Post, J. B. et al., Nephron Clin. Pract. 116, c247–c255 (2010) or Schneider, S. M. et al. Nephrol. Dial. Transplant. 30, 1551–1559 (2015). Are there possibly protective effects of exercise if no improvement has already been seen?

- Comparison of the findings with the healthy population is missing.

Author Response

Grigoriou et al. Healthcare 2024

We would like to sincerely thank both reviewers for their constructive review and useful comments on the manuscript. Our manuscript has been revised in a point-by-point fashion according to the suggestions of the editorial board and the reviewers.

We believe that the manuscript is now much more precise and more complete. We thank the reviewers for their productive comments and suggestions, and we hope that these revisions now make the manuscript suitable for publication in Healthcare.

Yours sincerely,

Giorgos K. Sakkas (GKS) (on behalf of all authors)

Reviewer#1

The their study, the authors investigated the influence of a nine month bicycle ergometer training during hemodialysis sessions 3 times a week on emotional intelligence and health related quality of life.

Initially 78 patients were screened, 23 completed the subsequent 9-month training. Patients with poor initial results in EI tests improved in terms of quality of life and fatique, patients with better initial results did not improve as a consequence of the intervention.

The question posed in this study is very important in the context of the care of patients on hemodialysis and the current issue of the importance of sport on dialysis and covers a new aspect that has not been investigated before. The type of training and the duration of the study seem reasonable.

GKS: We thank the reviewer for the supportive comments.

However, the article raises some questions: 

Methods/Design:

- What was the rationale for choosing the two EI tests? Why were common tests such as the Mayer-Salovey-Caruso Emotional Intelligence Test (MSCEIT) not used?

GKS: We thank the reviewer for the comments. For our study, we wanted to focus on both a “Self reported” test such as the  SSEIT as well as to assess EI from an external examiner such as the WLEIS. For both cases however, we wanted to assess EI as trait and not as ability. In addition, we wanted to distinct from the concept of the Five Personalities Model, something which is represented or expressed better by MSCEIT. Since we wanted to test the effect of chronic exercise on EI (as trait), we believed that those 2 questionnaires will give us the full range of tools to investigate EI.

- What is the rationale for the division into 4 subgroups depending on the test procedure? This should be explained in more detail.

GKS: We apologize for not clearly describing the study’s hypothesis. The rationale behind the 2 WLEIS and 2 SSEIT subgroups is related to the fact that patients with low or high values of EI could respond differently into various treatments or approaches, as we have seen in various other conditions. For example, patients with high scores in depression symptoms respond better to CBT or to medication compared to patients with mild symptoms and vice versa. Having that in mind, we tested the hypothesis that the level of EI could affect responsiveness to exercise training.  

The above information has been added to the revised version of the manuscript as appropriate. [Line 79] [In addition, the rationale behind the development of 4 patient subgroups based on WLEIS and SSEIT scores is to assess whether the level of EI (Low or High score) could affect the effectiveness of an exercise training program.]

-How are the results to be evaluated in comparison to the healthy normal population? The authors should address this in the discussion section.

GKS: We thank the reviewer for pointing this out. According to the literature, high physical activity levels have been associated with higher EI levels in healthy individuals. This has been included in our introduction. [Line 65] [According to the current literature, high physical activity levels are associated with good levels of EI in healthy individuals [16, 17], while it is still unknown whether exer-cise can improve EI in hemodialysis patients.]

- Why were the tests in the non-intervention group not also repeated after 9 months? Does repetitive testing play a role with these tests?

GKS: We thank the reviewer for pointing this out. Unfortunately, the non-intervention group did not agree to participate in any other measurements and therefore, we could not acquire those data.  

Results: 

- Table 3: Number of patients for SSEIT and WLEIS does not add up to 78, please explain. 

GKS: We thank the reviewer for pointing this out. The reason for that is that the groups were developed using the quartile approach, excluding the median quartile values (patients with middle scores). Therefore, the number of participants is less. We have included this in our methodology as follows: [Line 82] [The four patient subgroups were developed using the quartile approach, using the lower quartile for the Low group and the upper quartile for the High group, excluding the values for the median quartile.]

- Table 5: Do the p-values refer to the differences between the categories? There is comment regarding the significance of the “delta change”. The "Low" category consists of only 3 patients. How does this influence the results? Please discuss.

GKS: We thank the reviewer for pointing this out. The “P value” is refered to the GLM repeated measures between the categories. Only Delta-changes were statistically significant. The * symbol refers to within-group deferences (pre-post). Only the low EI group showed significant differences. Indeed, the N=3 patients is problematic, and this is discussed in the weakness section. [Line 399] [The small pool of subjects that completed the exercise intervention, the N=3 in one of the subgroups (Low EI), and the lack of a control group could have jeopardized the generalization of the findings of the current study.]

Discussion:

- The part of the discussion on the influence of chronic diseases on EI and vice versa takes up a long section and could be shortened. 

GKS: We thank the reviewer for this suggestion. This part has been shortened as suggested.

- Line 350: The statement is contrary to the previous statement that training at low initial values has an influence. 

GKS: We thank the reviewer for pointing this out. The text now reads as follows: [Nine months of exercise training improved the EI status of the HD patients with low WLEIS and not in the whole sample. The failure of the examined exercise intervention to improve the EI status of the whole sample These results could be attributed to the fact that the exercise intensity was moderate, between 50-60% of the patient’s maximal exercise capacity, while a higher level of intensity could have been more effective.’]

- Training at 50% of maximal exercise capacity for 1 h is above the usual training levels in other studies. This should be compared, e.g. reference to the recently published DiATT study (NEJM Evid 2023;2(9)). 

GKS: According to published guidelines on exercise in hemodialysis patients, aerobic exercise at an intensity above 60% of the patients’ maximum capacity is recommended to improve various physiological and health-related parameters, which is why we chose the intensity for the current study. Nevertheless, we acknowledge that this level of intensity may vary compared to the levels utilized in similar studies within the same population. Based on the literature, aerobic exercise training at 50-60% of maximal exercise capacity is considered to be moderate-intensity exercise.

We added the above information to the discussion section as appropriate. [Line 365] [According to published guidelines on exercise in hemodialysis patients, aerobic exercise at an intensity above 60% of the patients’ maximum capacity is recommended to improve various physiological and health-related parameters (Smart et al. 2013), which is why we chose the intensity for the current study. However, we are aware that this level of intensity is higher (Anding-Rost et al. 2023)  or lower (Morais et al. 2019) compared to the levels utilized in similar studies within the same population. Based on the literature, aerobic exercise training at 50-60% of maximal exercise capacity is considered to be moderate-intensity exercise.]

- The authors should make a statement to the usually worsening cognitive function in chronic hemodialysis patients (for example Post, J. B. et al., Nephron Clin. Pract. 116, c247–c255 (2010) or Schneider, S. M. et al. Nephrol. Dial. Transplant. 30, 1551–1559 (2015). Are there possibly protective effects of exercise if no improvement has already been seen?

GKS: We thank the reviewer for the suggestion and the references provided. This statement has been added to the revised version of the manuscript as appropriate. [Line 381] [Cognitive impairment is common in chronic kidney disease patients, particularly among those receiving hemodialysis therapy (Post et al. 2010; Schneider et al. 2015). Cognitive impairment is common in chronic kidney disease patients, particularly among those receiving hemodialysis therapy [46, 47]. Although exercise intervention was found to be beneficial specifically for patients with low EI levels in the present study, prior research has demonstrated the effectiveness of intradialytic exercise in enhancing cognitive function [48]. The increased cerebral blood flow resulting from exercise has been identified as a potential mechanism through which intradialytic aerobic exercise may help reduce cognitive decline and preserve brain function [49]. Other potential mechanisms include improved sleep, reduced depression, and improved cardiovascular-related parameters, all reported to be improved after intradialytic aerobic exercise training in HD populations [50, 51]. Unfortunately, the investigation of the physiological mechanisms that could explain the beneficial effects of exercise on EI and cognition was out of the scope of this trial.

- Comparison of the findings with the healthy population is missing.

GKS:  We thank the reviewer for pointing this out. Unfortunately, no specific articles assess exercise training interventions and EI in healthy individuals, while only a few discuss the relationship between physical activity levels and EI. We have included a statement in the introduction section. [Line 65] [According to the current literature, high physical activity levels are associated with good levels of EI in healthy individuals [16, 17], while it is still unknown whether exercise can improve EI in hemodialysis patients.]

Reviewer 2 Report

Comments and Suggestions for Authors

In this manuscript, Grigoriou et al. provides valuable insights into the relationship between emotional intelligence (EI), functional capacity, fatigue, cognitive function, and quality of life (QoL) in HD patients. I only have minor comments:

- The overall sample size (78 patients at first and 18 patients that completed the exercise training program) could be considered relatively small.

- The gender distribution in the subgroup completing the exercise program is imbalanced (15M/3F). Is this imbalance due to gender distribution in hemodialysis patients?

- It is interesting that nine months of exercise training is able to improve EI scores in patients with low EI levels. But it needs to be interpreted cautiously. Please include further discussion on potential underlying mechanisms?

- It would be beneficial to discuss potential confounding factors or limitations that could have influenced the results, such as comorbidities, or medication use that participants may have undergone during the study period.

- Please add potential implications of the findings for clinical practice or future research directions, such as exploring tailored interventions based on EI levels after the conclusions?

Author Response

Grigoriou et al. Healthcare 2024

We would like to sincerely thank both reviewers for their constructive review and useful comments on the manuscript. Our manuscript has been revised in a point-by-point fashion according to the suggestions of the editorial board and the reviewers.

We believe that the manuscript is now much more precise and more complete. We thank the reviewers for their productive comments and suggestions, and we hope that these revisions now make the manuscript suitable for publication in Healthcare.

Yours sincerely,

Giorgos K. Sakkas (GKS) (on behalf of all authors)

Reviewer#2

In this manuscript, Grigoriou et al. provides valuable insights into the relationship between emotional intelligence (EI), functional capacity, fatigue, cognitive function, and quality of life (QoL) in HD patients. I only have minor comments:

GKS: We thank the reviewer for the supportive comments.

- The overall sample size (78 patients at first and 18 patients that completed the exercise training program) could be considered relatively small.

GKS: We thank the reviewer and we totally agree. The small sample size of the current study has been added to the limitations section of the manuscript as appropriate. [Line 403] [ Finally, as the intervention part of the study used a small number of patients, the results should be treated with caution.]

- The gender distribution in the subgroup completing the exercise program is imbalanced (15M/3F). Is this imbalance due to gender distribution in hemodialysis patients?

GKS: We thank the reviewer for pointing this out. Indeed, in our study, there is an imbalance between males and females. This is also a common issue in many trials as it has been reported recently in a systematic review (Soomro et al. 2023).The reason of this imbalance is related to the population of our dialysis unit. Indeed, this balance is evident in our cohort with 50 males and 28 females patients. We have included it in our weaknesses section. [Line 402] [In addition, the cohort of patients was not gender balanced as it can be seen in both cross-sectional and intervention parts of the study.]

- It is interesting that nine months of exercise training is able to improve EI scores in patients with low EI levels. But it needs to be interpreted cautiously. Please include further discussion on potential underlying mechanisms?

GKS:Indeed, due to the fact that our interventional group was very small, our findings should be interpreted with caution. We don’t have a clear picture of how and why exercise training improved EI, however, some evidence supports the idea that exercise and EI are working complementary enhancing the patients' inner world with a better resilience to cope with the outside world involving and engaging a higher cognitive function such as attention, memory, regulation, reasoning, awareness, monitoring, and decision-making (SzczygieÅ‚ D., Mikolajczak M. Personal. Individ. Differ. 2017).

- It would be beneficial to discuss potential confounding factors or limitations that could have influenced the results, such as comorbidities, or medication use that participants may have undergone during the study period.

GKS: We thank the reviewer for the suggestion. We do have the comorbidities and medication, however, we believe that the study could become chaotic after all this information. We have not detected any potential confounder to the EI score.

- Please add potential implications of the findings for clinical practice or future research directions, such as exploring tailored interventions based on EI levels after the conclusions?

GKS: We thank the reviewer for the suggestion. We have included the following statement at the end of the discussion section. [Line 393] [ Emotional intelligence is a promising parameter for influencing physical and psychological characteristics in populations suffering from chronic conditions or diseases. It seems that exercise, especially chronic exercise training, can alter the state of EI, improving the mental state and facilitating changes in the physical component. However, more research is necessary to assess the precise exercise prescription needed for maximizing the positive effect on human health.]

Round 2

Reviewer 1 Report

Comments and Suggestions for Authors

Study description and discussion improved significantly. Thank you.

Reviewer 2 Report

Comments and Suggestions for Authors

The revised manuscript looks fine to me. My concerns have been solved.